# Lack of Evidence of Hepatitis E Virus Infections in a Cohort of Boars and Deer Species in a Game Reserve in Northern Germany

Tim Westphal [1], Michel Delling [2], Maria Mader [1], Christin Ackermann [1], Thomas Horvatits [1], Marc Lütgehetmann [3], Julian Schulze zur Wiesch [1,4], Sven Pischke [1,4,*,†] and Claudia Beisel [1,4,5,†]

1 Department of Medicine, University Medical Center Hamburg-Eppendorf, 20251 Hamburg, Germany
2 Revierförsterei Klövensteen, Sandmoorweg 150, 22559 Hamburg, Germany
3 Center for Diagnostics, Institute of Medical Microbiology, Virology and Hygiene, University Medical Center Hamburg-Eppendorf, 20251 Hamburg, Germany
4 German Center for Infection Research (DZIF), Lübeck—Borstel—Riems, 20251 Hamburg, Germany
5 Department of Internal Medicine IV, Gastroenterology and Infectious Diseases, University Hospital Heidelberg, 69120 Heidelberg, Germany
* Correspondence: s.pischke@uke.de
† These authors contributed equally to this work.

**Abstract:** The risk of acquiring hepatitis E virus (HEV) infections by wild animals living in the European wild nature has previously been reported and high anti-HEV antibody detection rates were detected in several animal species. However, data on the HEV seroprevalence of wild boars and deer held in game reserves are rare. In the present study, we investigated anti-HEV seroprevalence and HEV RNA in 38 deer and 15 wild boars living in a game reserve in Northern Germany. Surprisingly, none of the animals tested positive for HEV RNA in blood, liver, or muscle (diaphragm), and all animals (n = 53, 100%) were anti-HEV negative. In conclusion, HEV infections in enclosed areas, such as game reserves, in Germany are rare, and the risk of HEV transmission through meat from these animals to humans seems to be low.

**Keywords:** HEV infection; HEV seroprevalence; wildlife; game reserve

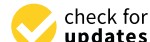



## 1. Introduction

Hepatitis E virus (HEV) infection in humans is mainly characterized by acute liver disease. In most cases, HEV infection is a self-limiting hepatitis and many cases are asymptomatic. However, in immunocompromised patients, there is a risk of chronic courses, which may lead to liver cirrhosis, with life-threatening complications over a few years [1].

There are eight HEV genotypes within *Paslahepevirus balayani* (*Hepeviridae* family) [2]. Genotypes 1–4 are implicated in human infection, while genotypes 5 and 6 are restricted to wild boars [3,4], and genotypes 7 and 8 have been described to be able to infect camels and in rare cases humans [5]. Genotypes 1 and 2 mainly affect humans through faecal–oral transmission in tropical, developing countries. Genotypes 3 and 4—which are the most common genotype in Western countries—have frequently been detected in humans, as well as in various other animal species, including swine and deer [1]. Domestic pigs and wild boars represent the main reservoirs for these genotypes and thus have a high prevalence of anti-HEV antibodies [6]. In Germany, anti-HEV seroprevalence rates of about 33% in wild boars [7] and 50% in domestic pigs have been described [8]. Adloch et al. revealed an even higher prevalence of HEV-ribonucleic acid (RNA) positivity in German wild boars (68%) [9]. In other studies, wild ruminants are assumed to be involved in the transmission cycle of HEV [6,10–13]. Nonetheless, deer do not seem to be a major reservoir of HEV, but rather their HEV infection coincides with the physical closeness to home ranges of wild boars that excrete HEV to a great extent by faeces [6,14]. In contrast to free wild animals, few data are available for anti-HEV seroprevalence and HEV-RNA analysis of different

wild animal species living in enclosures, which means that they are separated from HEV transmitters, such as wild boars, by fences [15,16]. However, an HEV transmission still remains conceivable through the snout, saliva, and faeces of external wild animals flanking the fence.

The results of this study further contribute to the knowledge of the epidemiology of HEV. Given the limited amount of data available on the susceptibility to HEV infection of wild animals living in game reserves, this work will be of interest to the HEV epidemiology field; but, it cannot be used to help to assess the overall risk of zoonotic transmission of HEV from wild animals living in enclosures to the human population. The overall aim of this study was to determine the HEV seroprevalence of boars and deer in a German game enclosure to assess the risk of zoonotic HEV transmission for human visitors of enclosures and consumers of meat from wild boars or deer living in enclosures.

## 2. Methods

### 2.1. Animals

All animals studied were born and raised in the *Wildgehege Klövensteen* game reserve, so that the date of their birth and thus their age at the time of the study were known.

### 2.2. Sampling

Blood and tissue samples from the muscle (diaphragm) and liver of wild boars and deer (n = 53) were collected at the *Wildgehege Klövensteen*, a game reserve located at the northwestern boundary of the federated state of Hamburg, Germany. The samples were collected during December 2020 from fallow deer (*Dama Dama*) (n = 24, 45%), red deer (*Cervus elaphus*) (n = 14, 27%), and wild boars (*Sus scrofa*) (n = 15, 28%) during the culling arranged for the control and management of wild boar and deer in the game reserve. While blood and liver samples were available from all animals (n = 53, 100%), muscle samples were available for only 46 animals (87%) (Table 1).

**Table 1.** Animal details and serological prevalence of HEV in wild animal samples from *Klövensteen* (game reserve).

| | All Animals (n = 53, 100%) | Red Deer (n = 14, 27%) | Fallow Deer (n = 24, 45%) | Wild Boar (n = 15, 28%) |
|---|---|---|---|---|
| **Sex** | | | | |
| male, n (%) | 21 (40) | 5 (36) | 9 (37,5) | 7 (47) |
| female, n (%) | 32 (60) | 9 (64) | 15 (62,5) | 8 (53) |
| **Age**, in years | | | | |
| <1 | 24 (45) | 4 (29) | 5 (21) | 15 (100) |
| 2–4 | 19 (36) | 2 (14) | 17 (71) | 0 (0) |
| >4 | 9 (17) | 7 (50) | 2 (8) | 0 (0) |
| >10 | 1 (2) | 1 (7) | 0 (0) | 0 (0) |
| **HEV IgG**, serum | | | | |
| available, n (%) | 53 (100) | 14 (100) | 24 (100) | 15 (100) |
| negative, n (%) | 53 (100) | 14 (100) | 24 (100) | 15 (100) |
| positive, n (%) | 0 (0) | 0 (0) | 0 (0) | 0 (0) |
| **HEV RNA**, serum | | | | |
| available, n (%) | 53 (100) | 14 (100) | 24 (100) | 15 (100) |
| negative | 53 (100) | 14 (100) | 24 (100) | 15 (100) |
| positive | 0 (0) | 0 (0) | 0 (0) | 0 (0) |
| **HEV RNA**, muscle | | | | |
| available, n (%) | 46 (87) | 9 (64) | 22 (92) | 15 (100) |
| negative | 46 (100) | 9 (100) | 22 (100) | 15 (100) |
| positive | 0 (0) | 0 (0) | 0 (0) | 0 (0) |
| **HEV RNA**, liver | | | | |
| available, n (%) | 53 (100) | 14 (100) | 24 (100) | 15 (100) |
| negative | 53 (100) | 14 (100) | 24 (100) | 15 (100) |
| positive | 0 (0) | 0 (0) | 0 (0) | 0 (0) |

### 2.3. Serological Assay

All serum blood samples were analysed for HEV-specific immunoglobulins (IgG and IgM) with two different veterinary ELISA kits according to the manufacturer's instructions (MP Biomedicals Germany GmbH, Eschwege, Germany and PrioCHECK™ Porcine HEV Ab Strip Kit, Thermo Fisher Scientific GmbH, Dreieich, Germany). In short, a microplate pre-coated with a soluble HEV antigen labeled with horseradish peroxidase was added together with the specimen to the test. After incubation for 60 min at 37 °C and a washing step, the plate was incubated with a substrate solution (tetramethylbenzidine) for 30 min at 37 °C in the dark. The reaction was stopped with a sulphuric acid solution and the absorbance was determined at a 450 nm wavelength using a microplate reader (FLUOstar omega, BMG LABTECH GmbH, Ortenberg, Germany).

### 2.4. Molecular Assay

Blood, liver, and muscle tissue specimens were tested for HEV RNA by real-time quantitative polymerase chain reaction (rtPCR) at the Institute of Medical Microbiology, Virology and Hygiene of the University Medical Center Hamburg-Eppendorf by the Cobas Taqman 6800 (Roche, Hilden, Germany), according to the manufacturer's instructions [17].

### 2.5. Ethics

The study was conducted in accordance with the German animal protection act. Due to the local regularizes and the German law [18], an ethical court vote was not necessary for this study as no human samples were included.

### 3. Results

There was a higher proportion of female animals (n = 32, 60%). Twenty-four animals were up to 1 year old, 19 were between 2 and 4 years old, and 9 were above 4 years old, with only one red deer being older than 10 years.

The presence of HEV antibodies was tested in all animals (n = 53; 100%), using the MP Elisa veterinary kit (MP Biomedicals Germany GmbH, Eschwege, Germany), which is designed for veterinary blood sample analysis. Anti-HEV antibodies could not be detected in any animal. To confirm the results, we decided to run a second analysis with the PrioCHECK HEV Antibody ELISA; thus, all serum samples were tested twice. Again, no anti-HEV antibodies could be detected.

To rule out early ongoing HEV infection, HEV rtPCR was performed in different tissue samples. HEV RNA could not be detected in any of the respective samples (n = 0; 0%), excluding active HEV infection in all of the 53 animals (Table 1).

Taken together, the overall HEV-IgG seroprevalence of the presented animal cohort was 0%.

A total number of 53 animals were included in this study: 14 red deer (27%), 24 fallow deer (45%), and 15 wild boars (28%). There were 32 female animals (60%). Samples were collected from serum samples, muscle, and liver tissue. All animals tested negative for serum HEV IgG (n = 53, 100%). HEV RNA was detected by rtPCR in none of the samples (n = 0, 0%).

### 4. Discussion

Domestic pigs and various wild animal species, especially wild boars, represent an HEV reservoir. Two recent German studies also identified wild deer as a potential HEV reservoir by detecting HEV-specific antibodies and HEV RNA in a small number of red deer, roe deer, and fallow deer species [14,19]. Game enclosures try to avoid resident animals getting in contact with infectious wild-life animals, including HEV-infected excrements from other species by fencing. Thus, these game reserve animals live in an isolated cohort. However, the possible relevance of zoonotic transmission from wild animals to humans has been demonstrated several times as professionals, including forest workers and hunters, have higher HEV seroprevalence rates than the general population [20,21]. The present

study demonstrates that game reserve animals, at least in Northern Germany, do not relevantly contribute to this zoonotic risk.

An HEV variant distinct from the classical HEV genotypes, the so-called rat HEV, attracted a lot of attention in recent years. However, the role of HEV transmission by rats or other small rodents still needs to be determined [22,23]. Whether animals living in enclosures are at increased risk of HEV infection and the consumption of raw game meat is associated with HEV exposure so far has been insufficiently studied.

To make a further contribution to the study of the zoonotic transmission of HEV from animals bred in game reserves to humans, we evaluated the anti-HEV seroprevalence of wild animals living exclusively in "*Wildgehege Klövensteen*", a game enclosure in Hamburg, Germany. The overall HEV IgG seroprevalence of the analysed animal cohort was 0%. In general, the applied ELISA methods have to be considered to generate reliable results, which allows the authors to conclude that no recent or prior HEV infection could be identified in the analysed animal cohort. Recently acquired, acute HEV infection was excluded in all animals by negative rtPCR in serum samples, muscle, or liver tissue. Unfortunately, we were not able to collect stool samples. As HEV shedding in the stool might outlast presence in the blood stream for more than one month, it might be interesting to collect stool samples longitudinally in animal cohorts in future studies. However, particles in the stool originate from the liver and, in the present study, all tested livers were PCR negative. The absence of HEV infections in deer and wild boars living in enclosures in this study may be explained by the successful separation of wild animals from outside of the enclosure. Consistent with the presented data, a recent study from the Czech Republic detected no HEV RNA in fallow deer living in enclosures [24]. In contrast, wild boars of these Czech Game enclosures were HEV RNA positive in 23% (n = 61/260). This deviation from the presented study may be explained by the small number of wild boars investigated. Recently, Trojnar et al. investigated 108 blood and 106 liver samples from fallow deer, red deer, and sika deer of game enclosures from 11 farms in Thuringia, Germany, for HEV infection [15]. HEV RNA was detected in none of the animals. Only 4% of the serum samples were scored borderline for HEV-specific antibodies in the cohort of the fallow deer (n = 3/73). These findings further support the results that indicate a very low HEV seroprevalence in wild animals, including wild boars, in game reserves in Germany.

Taken together, the results of this study suggest a low risk of HEV infection in deer and wild boars living in a game enclosure in Northern Germany. However, the study was limited by a relatively small sample size. Future multicentric studies need to evaluate more wild animals living in game reserves from different geographic regions and other countries, in order to strengthen these findings. Furthermore, future studies should also include a longitudinal collection of stool samples from these animals to possibly detect transient carriership of the virus without seroconversion.

**Author Contributions:** Conceptualization: S.P. and M.D.; resources: S.P., M.D., T.W. and M.M.; data curation: C.B., T.W., C.A., T.H., M.L. and M.M.; data analysis: M.D., C.B. and M.L.; writing—review and editing: T.W., T.H., J.S.z.W., C.B. and S.P. All authors have read and agreed to the published version of the manuscript.

**Funding:** C.B. was funded by the DZIF Clinical Leave and Maternity Leave Program, Grant Number DZIF TI 07.005. S.P. received funding from the DZIF, Grant Number DZIF TI 07.001, and the Else-Kröner-Fresenius-Stiftung, Grant Number 2019: EKFS10.

**Institutional Review Board Statement:** The authors confirm that the ethical policies of the journal have been adhered to. No ethical approval was required as this study was conducted without human samples. The study was conducted in accordance with the German animal protection act. No animal was killed as a result of the study. All samples were collected after the animals' death due to planned shootings to keep the population limited. We reported the study to the responsible Department of Veterinary Affairs, Hamburg, Germany. The Department expressed no ethical concerns and approved the study (date of approval 2020/09/15). No human samples were included.

**Informed Consent Statement:** Not applicable.

**Data Availability Statement:** Data storage is performed by the Universitätsklinikum Hamburg-Eppendorf. Data are available upon request from the corresponding author and can be shared after confirming that data will be used within the scope of the originally provided informed consent.

**Conflicts of Interest:** The authors declare no conflict of interest.

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
