# Peer review of "Lack of Evidence of Hepatitis E Virus Infections in a Cohort of Boars and Deer Species in a Game Reserve in Northern Germany"

_zoonoticdis, doi:10.3390/zoonoticdis2040018_

Round 1

Reviewer 1 Report

The paper entitled “Lack of evidence of hepatitis E virus infections in a cohort of boars and deer species in a game reserve in Northern Germany” by Westphal T. et al., investigated anti-HEV seroprevalence and HEV RNA in deer and wild boars living in a game reserve in Northern Germany.

Even if the LINES’ NUMERATION IS MISSING AND IT IS DIFICULT TO INDICATE THE REVIEWS POSITION, the reviewer highlighted editing sentences and inserted the comments into the downloaded .pdf original file, that is going to be re-sent to the journal.

The Abstract, Introduction, Methods and Results are sufficiently written even to be improved; the Discussion would need to be enlarged and, to be clearer, especially in the final part. In fact, it seems an ensemble of copied-pasted sentences without a linearity in the content.

In summary, the manuscript could be accepted, should the authors take in consideration some major and minor comments to improve the quality of the work.

Major revisions:

·         this manuscript is missing the number of lines (rows) and it is very difficult for the reviewer to indicate what and where edit the text!!

·         Intro: the authors should specify if these enclosures are actually separated by the rest of territory: even in the presence of a fence the virus can be transmitted through the snout, saliva and feces of external wild animals flanking the fence.

·         Page 2: why the authors write "farmed" if the manuscript is on wild animals?

·         Paragraphe 2.1: the authors should put the scientific name in latin (written in italic) of deer and w. boars

·         Paragraphe 2.3: Molecular detection of HEV-RNA in blood and muscle is very difficult to test as positive, since it is possible only during the period of viremia, that is a small window lasting 1 week or little more, while virus shedding by feces may persist up to 7 weeks after. The authors should have collected feces to monitor the virus, even to detect it before its arrival to the liver.

·         Capt. 3 Results. Starting phrase: this is the final part of the Introduction: move above. However, it is unclear the link between wild animals living in a restricted (fenced???) game area and farmed animals and humans. Are there any pig farms in the game area? It is not specified.

·         Page 3: this sentence must be moved to the Discussion. It remains, however, to be sure that the sensitivity and the specificity of the tests are high enough. Also, do not use "our" but be impersonal (...allow the authors to conclude....) trhough all the manuscript.

·         it still remains unclear why the authors do not clarify the connection between wild animals, humans and domestic animals.

·         It is very unclear if animals living in game areas are considered farmed or still wild!!

Minor revisions:

See the .pdf edited sent file rev2.

Author Response

Subject: Re-submission of the revised original research article entitled “Lack of evidence of hepatitis E virus infections in a cohort of boars and deer species in a game reserve in Northern Germany. by Westphal et al.

Dear Prof. Dr. Wikel,

We thank you for providing us the opportunity to re-submit a revised version of our manuscript entitled Lack of evidence of hepatitis E virus infections in a cohort of boars and deer species in a game reserve in Northern Germany.” by Westphal et al. for consideration for publication in Zoonotic Diseases.

We greatly appreciate the constructive and timely feedback from you and the reviewers and believe that these revisions have further strengthened our manuscript. We hope that the revised manuscript is acceptable for publication in its current form.

Please find enclosed our detailed point-by-point responses to the reviewer’s comments and how we have specifically addressed each of his/her concerns. All changes in the manuscript are marked in yellow.

With best regards,

Tim Westphal, PD Dr. Sven Pischke and Dr. Claudia Beisel

(on behalf of all authors)

Detailed Response to Reviewers

Reviewer 1:

“The paper entitled “Lack of evidence of hepatitis E virus infections in a cohort of boars and deer species in a game reserve in Northern Germany” by Westphal T. et al., investigated anti-HEV seroprevalence and HEV RNA in deer and wild boars living in a game reserve in Northern Germany.

Even if the LINES’ NUMERATION IS MISSING AND IT IS DIFICULT TO INDICATE THE REVIEWS POSITION, the reviewer highlighted editing sentences and inserted the comments into the downloaded .pdf original file, that is going to be re-sent to the journal.

The Abstract, Introduction, Methods and Results are sufficiently written even to be improved; the Discussion would need to be enlarged and, to be clearer, especially in the final part. In fact, it seems an ensemble of copied-pasted sentences without a linearity in the content.

In summary, the manuscript could be accepted, should the authors take in consideration some major and minor comments to improve the quality of the work.

We thank the Reviewer for the overall positive assessment of the presented data and the thoughtful comments raised. Please find enclosed our detailed point-by-point responses and how we have specifically addressed your concerns

Major revisions:

  1. This manuscript is missing the number of lines (rows) and it is very difficult for the reviewer to indicate what and where edit the text!!

We sincerely apologize and regret that the number of lines was not visible to you. Our submitted version included the line numbering, so unfortunately something must have gone wrong in the editing process. We now again included the line numbering and hope it appears to you.

  1. Intro: the authors should specify if these enclosures are actually separated by the rest of territory: even in the presence of a fence the virus can be transmitted through the snout, saliva and feces of external wild animals flanking the fence.

We thank the Reviewer for highlighting this important aspect. We now revised the introduction section. This part now reads:

“In contrast to free wild animals, few data are available for anti-HEV seroprevalence and HEV-RNA analysis of different wild animal species living in enclosures, which means that they are separated from HEV-transmitters such as wild boars by fences [15, 16]. Even though a HEV transmission still remains conceivable through the snout, saliva and feces of external wild animals flanking the fence.”

Furthermore, we added “separated by fences” (line 59) to the main text to clarify this aspect.

  1. Page 2: why the authors write "farmed" if the manuscript is on wild animals?

We agree with the reviewer that we do not analyze the HEV prevalence in farmed animals and now consequently change this term throughout the manuscript.

  1. Paragraphe 2.1: the authors should put the scientific name in latin (written in italic) of deer and w. boars.

We thank the reviewer for this advice and now added the scientific names in latin. The sentence now reads:

“The samples were collected during December 2020 from fallow deer (Dama Dama) (n=24, 45%), red deer (Cervus elaphus) (n=14, 27%), and wild boars (Sus scrofa) (n=15, 28%) during the culling arranged for the control and management of wild boar and deer in the game reserve.”

  1. Paragraphe 2.3: Molecular detection of HEV-RNA in blood and muscle is very difficult to test as positive, since it is possible only during the period of viremia, that is a small window lasting 1 week or little more, while virus shedding by feces may persist up to 7 weeks after. The authors should have collected feces to monitor the virus, even to detect it before its arrival to the liver.

We thank the Reviewer for highlighting this important aspect. Unfortunately, we were not able to collect stool samples. As HEV shedding in the stool might outlast presence in the blood stream for more than one month, it might be interesting to collect stool samples longitudinally in animal cohorts in future studies. However, particles in the stool originate from the liver and in our study, all tested livers were HEV PCR negative. The absence of HEV infections in deer and wild boars living in enclosures in our study may be explained by the successful separation of wild animals from outside of the enclosure. We now addressed this important point in the discussion section of our manuscript.

  1. Capt. 3 Results. Starting phrase: this is the final part of the Introduction: move above. However, it is unclear the link between wild animals living in a restricted (fenced???) game area and farmed animals and humans. Are there any pig farms in the game area? It is not specified.

We thank the Reviewer for bringing this to our attention. We agree and now changed the manuscript accordingly.

  1. Page 3: this sentence must be moved to the Discussion. It remains, however, to be sure that the sensitivity and the specificity of the tests are high enough. Also, do not use "our" but be impersonal (...allow the authors to conclude....) trhough all the manuscript.

We thank the Reviewer for this valuable suggestion and now revised the manuscript respectively.

  1. it still remains unclear why the authors do not clarify the connection between wild animals, humans and domestic animals.

We thank the Reviewer for bringing up this important point for discussion. We now clarified this in the introduction section. This part now reads: “The overall aim of this study was to determine the HEV seroprevalence of boars and deer in a German game enclosure to assess the risk of zoonotic HEV transmission for human visitors of enclosures and consumers of meat from wild boars or deer living in enclosures.“

  1. It is very unclear if animals living in game areas are considered farmed or still wild!!

We thank the Reviewer for bringing this to our attention. We agree and now explicitly stated throughout the manuscript that we analyzed animals living in enclosures. We do not consider these animals to be wild or farmed.

We again thank the reviewer for her/his valuable suggestions and believe that these revisions have further strengthened our manuscript. We hope that the revised text now finds your approval.

Reviewer 2 Report

Westphal et al. were interested in the HEV infection in a game reserve in Northern Germany. A total of 53 animals, including 38 deer and 15 wild boars, were investigated for the presence and prevalence of anti-HEV antibodies and HEV RNA. Their results are highly compelling to me since two different ELISA kits were used for animal serum samples, and viral RNA was detected in the liver, blood, and muscle specimens of captured animals. Although none of those animals has shown any evidence of previous or ongoing HEV infection, the results still provide useful information on HEV infection in deer and wild boars in game reserves. After all, “negative” results also have positive meanings. Overall, the manuscript is very well-written, and the study design has been appropriately conducted. I only have minor comments that the authors may consider.

1.     Abstract: my primary concern is that the conclusion of the HEV infection in wild animals in enclosed areas is low, and the risk of HEV transmission to humans seems to be low, which seems over phrased. The results from only two game enclosures in Hamburg and Thuringia are hard to represent the whole situation in Germany. Moreover, as described by the authors, 23% (61/260) were HEV RNA positive in wild boars in the Czech Republic, which is comparatively a significant discrepancy. Finally, the HEV infection rates in wild animals in game reserves of other European countries are largely unknown. Therefore, I suggest softening the statement mentioned above.

2.     Introduction, para 2, line 1: the classification of HEV has recently changed, the Orthohepevirus A is now called balayani in the Paslahepevirus genus. Please see the 2021 release of the family Hepeviridae in ICTV. The authors may also refer to Viruses 2022 (PMID: 35632647).

3.     Introduction, para 2, line 7: to my understanding, rare evidence demonstrated that cats and dogs carry gt 3 and 4 HEV. The host tropism of HEV has recently been discussed in Curr Opin Microbiol 2021 (PMID: 32810801).

4.     Methods, 2.1: please specify the time of sample collection.

5.     Methods, 2.1, Table 1: how were the ages of wild animals determined?

6.      Methods, 2.3: the molecular assay of HEV-specific rtPCR should be referenced.

7.     Discussion, para 1, line 12: “mice” could be deleted.

8.     Discussion, para 2 and 3: as noted earlier, please revise the conclusion accordingly.

Author Response

Subject: Re-submission of the revised original research article entitled “Lack of evidence of hepatitis E virus infections in a cohort of boars and deer species in a game reserve in Northern Germany. by Westphal et al.

Dear Prof. Dr. Wikel,

We thank you for providing us the opportunity to re-submit a revised version of our manuscript entitled Lack of evidence of hepatitis E virus infections in a cohort of boars and deer species in a game reserve in Northern Germany.” by Westphal et al. for consideration for publication in Zoonotic Diseases.

We greatly appreciate the constructive and timely feedback from you and the reviewers and believe that these revisions have further strengthened our manuscript. We hope that the revised manuscript is acceptable for publication in its current form.

Please find enclosed our detailed point-by-point responses to the reviewer’s comments and how we have specifically addressed each of his/her concerns. All changes in the manuscript are marked in yellow.

With best regards,

Tim Westphal, PD Dr. Sven Pischke and Dr. Claudia Beisel

(on behalf of all authors)

Reviewer 2:

Westphal et al. were interested in the HEV infection in a game reserve in Northern Germany. A total of 53 animals, including 38 deer and 15 wild boars, were investigated for the presence and prevalence of anti-HEV antibodies and HEV RNA. Their results are highly compelling to me since two different ELISA kits were used for animal serum samples, and viral RNA was detected in the liver, blood, and muscle specimens of captured animals. Although none of those animals has shown any evidence of previous or ongoing HEV infection, the results still provide useful information on HEV infection in deer and wild boars in game reserves. After all, “negative” results also have positive meanings. Overall, the manuscript is very well-written, and the study design has been appropriately conducted. I only have minor comments that the authors may consider.

We thank the Reviewer for the overall positive assessment of our data and the thoughtful comments raised.

  1. Abstract: my primary concern is that the conclusion of the HEV infection in wild animals in enclosed areas is low, and the risk of HEV transmission to humans seems to be low, which seems over phrased. The results from only two game enclosures in Hamburg and Thuringia are hard to represent the whole situation in Germany. Moreover, as described by the authors, 23% (61/260) were HEV RNA positive in wild boars in the Czech Republic, which is comparatively a significant discrepancy. Finally, the HEV infection rates in wild animals in game reserves of other European countries are largely unknown. Therefore, I suggest softening the statement mentioned above.

We thank the Reviewer for bringing this to our attention. As suggested, we now edited our manuscript according to the advice of the Reviewer and have toned down our conclusions.

  1. Introduction, para 2, line 1: the classification of HEV has recently changed, the Orthohepevirus A is now called balayani in the Paslahepevirus genus. Please see the 2021 release of the family Hepeviridae in ICTV. The authors may also refer to Viruses 2022 (PMID: 35632647).

We thank you for noticing this and apologize for using the old term. We now revised the sentence and cited the recommended literature. This part now reads:

“There are eight HEV genotypes within this Paslahepevirus balayani (Hepeviridae family) [2].”

  1. Introduction, para 2, line 7: to my understanding, rare evidence demonstrated that cats and dogs carry gt 3 and 4 HEV. The host tropism of HEV has recently been discussed in Curr Opin Microbiol 2021 (PMID: 32810801).

We thank the Reviewer for this valuable suggestion. We now revised this sentence and included the important citation in our manuscript.

  1. Methods, 2.1: please specify the time of sample collection.

We thank the Reviewer for this important comment and now added the time of sample collection into the methods section.

  1. Methods, 2.1, Table 1: how were the ages of wild animals determined?

All animals studied were born and raised in the Wildgehege Klövensteen game reserve, so that the date of their birth and thus their age at the time of the study were known. We now included this information into the manuscript.

  1.  Methods, 2.3: the molecular assay of HEV-specific rtPCR should be referenced.

We thank the Reviewer for bringing this to our attention. We agree and now included the corresponding reference (DOI: 10.1016/j.jcv.2020.104525).

  1. Discussion, para 1, line 12: “mice” could be deleted.

We thank the reviewer for this comment and now deleted “mice” in this sentence.

  1. Discussion, para 2 and 3: as noted earlier, please revise the conclusion accordingly.

As suggested, we now edited our manuscript according to the advice of the Reviewer and toned down our conclusions. All changes in the manuscript are highlighted in yellow.

We again thank the Reviewer for her/his constructive feedback and believe that the changes made accordingly have substantially strengthened the manuscript, which we now hope finds your approval.
